# Study on Carbon Emission Characteristics and Emission Reduction Measures of Lime Production—A Case of Enterprise in the Yangtze River Basin

**Erxi Wu, Qiaozhi Wang** [ID]**, Lihua Ke \* and Guangquan Zhang**

School of Resource and Environment Engineering, Wuhan University of Science and Technology, Wuhan 430080, China; wuerxi1999@163.com (E.W.); wangqiaozhi@wust.edu.cn (Q.W.); zhanggq@wust.edu.cn (G.Z.)
\* Correspondence: kelihua@wust.edu.cn

**Abstract:** A scientific carbon accounting system can help enterprises reduce carbon emissions. This study took an enterprise in the Yangtze River basin as a case study. The accounting classification of carbon emissions in the life cycle of lime production was assessed, and the composition of the sources of carbon emission was analyzed, covering mining explosives, fuel (diesel, coal), electricity and high-temperature limestone decomposition. Using the IPCC emission factor method, a carbon life cycle emission accounting model for lime production was established. We determined that the carbon dioxide equivalent from producing one ton of quicklime ranged from 1096.68 kg $CO_2$ equiv. to 1176.96 kg $CO_2$ equiv. from 2019 to 2021 in the studied case. The decomposition of limestone at a high temperature was the largest carbon emission source, accounting for 64% of the total carbon emission. Coal combustion was the second major source of carbon emissions, accounting for 31% of total carbon emissions. Based upon the main sources of carbon emission for lime production, carbon emission reduction should focus on $CO_2$ capture technology and fuel optimization. Based on the error transfer method, we calculated that the overall uncertainty of the life cycle carbon emissions of quicklime from 2019 to 2021 are 2.13%, 2.07% and 2.09%, respectively. Using our analysis of carbon emissions, the carbon emission factor of producing one unit of quicklime in the lime enterprise in the Yangtze River basin was determined. Furthermore, this research into carbon emission reduction for lime production can provide a point of reference for the promotion of carbon neutrality in the same industry.

**Keywords:** lime; the Yangtze River basin; carbon emissions characteristics; life cycle; IPCC emission factor method; carbon emission reduction measures





## 1. Introduction

In recent years, climate anomalies have become more common [1]. The Paris Agreement called for limiting global warming efforts to 1.5 °C by the end of the century. At present, more than 130 countries have committed to net zero, of which the Republic of Suriname and Bhutan are the countries that have achieved net zero emissions [2]. China has issued a policy pledge to strive to peak carbon emissions by 2030 and achieve carbon neutrality by 2060. Furthermore, China has called for an 18% reduction in carbon emissions by 2025 compared with 2020 levels. Therefore, China's carbon emission reduction is urgent and difficult.

Lime is the primary component for the cement industry. It can be divided into quicklime, hard-burned lime, slaked lime and dolomite lime. In recent years, the global demand for lime has been increasing. In 2022, the world's total lime production was 430 million tons. China was the most influential country for lime production, accounting for 75.6% of the world's lime production, far more than the United States (4%), India (3.7%), Russia (2.6%) and other countries [3]. Lime production is an industrial process

with large energy consumption and carbon emissions [4]. The importance of carbon emissions throughout the life cycle cannot be ignored; this includes mining, transporting raw materials, producing lime and transporting lime to where it is used. Furthermore, lime is widely used in construction, metallurgy, pulp and paper, chemical manufacturing and other industries; the use of it can also lead to significant carbon emissions. Therefore, the carbon emissions of the Chinese lime industry should not be overlooked. Reducing the carbon emissions of the Chinese lime industry will help achieve the national target.

This study examined a limestone mining enterprise in the Yangtze River Basin of China (referred to as "Mine A"). Using the IPCC emission factor method, a carbon emission life cycle accounting model for quicklime production was constructed and used to explore the carbon emission distribution in each production stage. Then, key carbon emission sources were identified, and effective low-carbon production measures for the largest carbon emission factors were proposed. This study provided the actual data of the lime enterprise in the Yangtze River basin and determined the carbon emission factor of producing unit quicklime. More sustainable production strategies can be proposed based on the study. The results will provide references for scientific decision-making on the industry's development and carbon neutrality.

Section 2 is a literature review. Section 3 summarizes the methodology. Section 4 provides the analysis and results. Section 5 proposes low-carbon transformation measures and recommendations. Section 6 provides the conclusions.

## 2. Literature Review

### 2.1. Carbon Emission from the Lime Industry

Previous research has calculated carbon emissions from the lime industry in different countries. The scholars considered mining, crushing, screening and calcination as the boundary of carbon accounting. They calculated that the production of one ton of lime entailed the emission of around 1.2 t of carbon dioxide in Cuba [5], 0.98 t to 1.2 t of carbon dioxide in China [6–10] and 1.26 t of carbon dioxide in the EU [11] based on activity data and carbon emission factors. Furthermore, they analyzed the carbon emission structure of lime production and concluded that the majority of carbon dioxide emission occurs during the decarbonation of limestone to lime and remains from fuel burning. The difference in carbon dioxide emission from lime production in different countries lies in the uncertainty of activity data potentially caused by differential production efficiency, energy consumption efficiency and the selection of carbon emission factors.

The uncertainty of carbon emission factors will affect the accuracy of carbon dioxide emission calculation. Error propagation methods are often used to estimate the uncertainty of carbon emissions [12]. Shan et al. adopt error propagation in measurements to evaluate the uncertainty of carbon emissions over the life cycle of lime production. The results showed that the uncertainty was between 2.83% to 3.34%. The uncertainty was caused by lime carbon emission factors7. Zhang et al. used the error propagation method to analyze the uncertainty of carbon emissions from ships. They calculated that the uncertainty of carbon emissions from ships was 3.2%. They found that the measure to control this uncertainty was to update the measurement standards and improve the configuration of measuring instruments [12]. Wang et al. analyzed the uncertainty of greenhouse gas emissions from agricultural activities based on the error propagation method. The result of uncertainty was 45.47%; the reason for this uncertainty was that the studied case lacked actual emission factors in the studied area and chose general emission factors [13].

Through calculating the uncertainty of carbon emissions using error propagation methods, they can improve their understanding of their own situation, thus allowing more accurate planning and implementation of emission reduction measures.

### 2.2. Carbon Emission Reduction in the Lime Industry

Measures to reduce carbon in the lime industry primarily focused on the calcination process, including strengthening the heat insulation of the kiln, adopting cleaner fuel,

utilizing the waste heat of the kiln, adopting energy-saving environmental protection kilns and applying carbon capture and storage (CCS) technology6. Carbon dioxide from high-temperature decomposition of limestone can be decarbonized using CCS technologies. CCS technologies include pre-combustion capture, post-combustion capture and oxygen-enriched combustion [14]. Post-combustion capture, including physical/chemical absorption, membrane separation and physical/chemical adsorption, is considered to be more suitable for the lime industry [15,16]. Physical/chemical absorption solvents, such as monoethanolamine, are usually taken as the benchmark solvent for carbon dioxide removal, and the carbon dioxide removal rate can reach 90% [17,18]. Also, membrane separation has a good application potential in the cement industry. Hagg et al. and Baker et al. applied membrane separation to plant testing and demonstrated that it could remove 80% of carbon dioxide [19,20]. In addition, physical/chemical adsorption has been applied in the lime industry. A typical physical/chemical adsorption device is the rotary adsorber device, which mainly uses vacuum-temperature-concentration pressure variation to adsorb carbon dioxide and has been used in cement plants in Canada [21].

The second major source of carbon emission is the combustion of fuel. Fuel switching has also been proposed as one of the solutions for carbon reduction in the cement industry. Traditional fuels in the industry include coal, petroleum coke, petroleum and natural gas [22]. Alternative fuels, such as waste tires, refuse derived fuel and straws, have been used in the cement industry [23,24] and can potentially be applied in the lime industry.

Overall, CCS and alternative fuel technologies are the most common carbon reduction technologies in the research, and they can provide an important basis for the study of carbon emission reduction measures.

## 3. Materials and Methods

### 3.1. Determination of Accounting Boundary

The determination of system boundaries is the basis of carbon emission accounting [25] in the study of the life cycle carbon emission characteristics of quicklime, based on the actual process flow distribution aggregation and other research [26]. The life cycle of quicklime production is divided into three stages: open-pit mining, crushing and calcination. The carbon emission system boundaries include three stages of direct and indirect carbon emissions, excluding the office services in the mining area, as shown in Figure 1.

**Figure 1.** Lime life cycle carbon emission accounting boundary.

### 3.2. Carbon Emission Source Composition

From the three stages of open-pit mining, crushing and calcination, the carbon emission sources of each stage were identified. Open-pit mining stage: Explosion is prior to extraction. Ammonium nitrate explosives in explosion can produce a large amount of carbon dioxide. In addition, the power for mining equipment and transportation system mainly depends on the consumption of fossil fuels, especially diesel. Crushing stage: The jaw crusher reduces the limestone to the required particle size, which relies on electricity. Calcination stage: the calcination production line is composed of a vertical preheater, rotary kiln, vertical cooler and other equipment. All this equipment relies on electricity to operate. Furthermore, limestone needs to be decomposed under high-temperature conditions, and this heat generally comes from the combustion of coal.

### 3.3. Models

Greenhouse gas accounting methodologies can be divided into the IPCC emission factor method, the measured method and the mass balance of carbon method. The measured method has a high level of uncertainty and is not often used for carbon emission [27]. The material balance algorithm is based on the analysis of carbon material flow balance without considering the specific process producing the emissions. The method is based on calculating the difference between the carbon input and carbon output of the industry [28]. Compared with the previous two methodologies, the IPCC emission factor method is an internationally recognized carbon emission assessment methodology, which can be used to estimate emission data for each category of emission [29]. The IPCC emission factor method specifically provides the carbon emissions of the industrial process, which is helpful in identifying the key carbon emission sources and which makes it ideal for carbon emission accounting in our study. Based on the IPCC emission factor method from the 2019 Refinement to the 2006 IPCC Guidelines for National Greenhouse Gas Inventories [30], the carbon emission accounting model of quicklime is as follows:

$$E_t = E_e + E_d + E_c + E_p + E_h \tag{1}$$

where $E_t$ represents the carbon dioxide equivalent of the life cycle of quicklime, kg $CO_2$ equiv; $E_e$ is the carbon dioxide emissions generated by explosives, kg $CO_2$ equiv; $E_d$ is carbon dioxide emissions from diesel combustion, kg $CO_2$ equiv; $E_c$ is carbon dioxide emissions from coal combustion, kg $CO_2$ equiv; $E_p$ is carbon dioxide emissions indirectly generated by electricity consumption, kg $CO_2$ equiv; $E_h$ is carbon dioxide emissions from high temperature decomposition of limestone, kg $CO_2$ equiv.

(1)    Explosion

Ammonium nitrate explosives release a large amount of carbon dioxide when detonated. The carbon emission calculation formula is:

$$E_e = AD_e \times EF_e(CO_2) \tag{2}$$

where $AD_e$ is the annual consumption of explosives, t/a. $EF_e$ ($CO_2$) is the carbon emission factor, t/t.

(2)    Diesel combustion

Greenhouse gas emission accounting methods and reporting guidelines for land transportation enterprises (Trial) (Development and Reform Office Climate [2015]1722) showed that the greenhouse gases of road transportation enterprises using diesel as fuel include $CO_2$, $CH_4$ and $N_2O$.

$$E_{d-CO_2} = AD_d \times NC_d \times EF_d(CO_2) \tag{3}$$

$$E_{d-CH_4} = AD_d \times NC_d \times EF_d(CH_4) \tag{4}$$

$$E_{d-N_2O} = AD_d \times NC_d \times EF_d(N_2O) \tag{5}$$

where $AD_d$ is the annual consumption of diesel oil for mining equipment and transportation equipment, t/a; $NC_d$ represents diesel average low calorific value, GJ/t. $EF_d$ ($CO_2$) is carbon emission factor for diesel, t/GJ. $EF_d$ ($CH_4$) is $CH_4$ emission factor for diesel, t/GJ. $EF_d$ ($N_2O$) is $N_2O$ emission factor for diesel, t/GJ.

(3)    Coal combustion

It is used as a fossil fuel of rotary kilns and produces $CO_2$.

$$E_c = AD_c \times NC_c \times EF_c(CO_2) \tag{6}$$

where $AD_c$ is the annual consumption of rotary kiln bituminous coal, t/a. $NC_c$ represents coal's average low calorific value, GJ/t. $EF_c$ ($CO_2$) is the carbon emission factor for coal, t/GJ.

(4)    Purchased electricity

China's power industry is dominated by thermal power generation. Therefore, the power industry is the key area of indirect carbon emission.

$$E_p = AD_e \times EF_e(CO_2) \tag{7}$$

where $AD_e$ is annual purchased power, MWh/a. $EF_e$ ($CO_2$) is the annual average power supply emission factor of the Central China Power Grid, t/MWh.

(5)    High-temperature decomposition of limestone

The decomposition of carbonate at high temperatures is the main factor that produces carbon dioxide.

$$E_h = 0.97 \times EF_l(CO_2) \times (M_l + \frac{M_d \times C_l \times EF_l(CO_2)}{0.52} \frac{M_l}{M_l + M_c}) \tag{8}$$

where $EF_l(CO_2)$ is the $CO_2$ emission factor of quicklime, t/t. $M_l$ is annual quicklime production, t/a. $M_d$ is the amount of dust removal, t/a. $M_l$ is the amount of limestone in the rotary kiln, t. $M_c$ is coal for rotary kiln combustion, t/a. $C_l$ is the carbonate content in limestone, %.

The calculation of the carbon emission factor of limestone is complicated. Jiao et al. have studied the carbonate carbon emission factor of the cement industry [31]. The calculation formula is as follows:

$$EF_l(CO_2) = RC \times 44/56 + RM \times 44/40 \tag{9}$$

where $R_C$ is the content of CaO in limestone, %. $R_M$ is the content of MgO in limestone, %.

For other carbon emission factor data, refer to 'General Principles of Comprehensive Energy Consumption Calculation GBT2589-2020' and 'Guidelines for Provincial Greenhouse Gas Inventories'. The specific values are shown in Table 1.

**Table 1.** Low calorific value and carbon emission factors.

| Materials/Energy | Low Calorific Value (GJ/t) | Carbon Emission Factors | | |
| --- | --- | --- | --- | --- |
| | | EF($CO_2$) | EF($CH_4$) | EF($N_2O$) |
| Explosive | / | 0.26 t/t | / | / |
| Diesel | 42.65 | 0.07 t/GJ | $3.90 \times 10^{-6}$ t/GJ | $3.90 \times 10^{-6}$ t/GJ |
| Electricity | / | 0.84 t/MWh | / | / |
| Coal | 19.57 | 0.09 t/GJ | / | / |
| Quicklime | / | 0.71 t/t | / | / |

### 3.4. Uncertainty Analysis

The error transfer method can quantify the accuracy of carbon emission accounting. The procedure of the error transfer method is to first determine the uncertainty associated with each quantity [30]. The calculation formula is as follows:

$$U_i = \frac{2(C - B)}{A} \times 100\% \tag{10}$$

where $U_i$ is the uncertainty of class is carbon emission factors, %; A is the calculated value of this study; B is the minimum of the emission factor and C is maximum of the emission factor.

When the uncertainties are combined via addition or subtraction, the standard deviation of the sum is the square root of the sum of the squares of the standard deviations of the additions, where the standard deviation is expressed as an absolute value. According to this description, Formula (11) can be used to derive the uncertainty of the sum.

$$U_{total} = \frac{\sqrt{\sum_1^n (U_i X_i)^2}}{|\sum_1^n X_i|} \tag{11}$$

where $U_{total}$ is the percent uncertainty of the sum of all quantities; $X_i$ and $U_i$ are the percent uncertainty of the uncertainty and its associated quantities, respectively.

In order to verify the difference between the carbon emission factors of quicklime production in this study and those provided by the IPCC, the values of carbon emission factors in this study and the maximum and minimum of carbon emission factors provided by the IPCC are listed in Table 2.

**Table 2.** Carbon emission factors from our study and IPCC.

| Material/Energy | Carbon Emission Factors | | |
| :---: | :---: | :---: | :---: |
| | **Values** | **Maximum** | **Minimum** |
| Diesel (kg/GJ) | 70.00 | 74.80 | 72.60 |
| Electricity (t/MWh) | 0.84 | 1.09 | 0.79 |
| Coal (t/GJ) | 0.09 | 0.10 | 0.089 |
| Quicklime (t/t) | 0.71 | 0.75 | 0.74 |

### 3.5. Data Sources

The life cycle of quicklime includes three stages. These three stages of carbon emission are divided into explosives, fuel combustion (diesel, coal), electricity consumption and high-temperature limestone decomposition. Diesel consumption is mainly caused by mining loaders, excavators, bulldozers, explosive transporters and mining vehicles. The electricity consumption covers the crushing stage and the calcination stage. In July 2022, a 10-day investigation was carried out at Mine A in the Yangtze River basin to obtain the basic data on material and energy flows from 2019 to 2021 by the production technology department. From 2019 to 2021, the amount of limestone produced by Mine A was 142, 145 and 147 million tons per year, respectively. The output of quicklime and the material and energy consumption for the production of quicklime from 2019 to 2021 are shown in Table 3.

The output data in Table 3 show a steady upward trend. In 2021, with effective epidemic prevention and control measures in the Yangtze River basin area, there was a significant increase in quicklime production, with a growth rate of over 34% compared to 2020.

**Table 3.** The output of quicklime and the material and energy consumption for the production of quicklime from 2019 to 2021.

| Producing Stage | Materials/Energy | 2019 | 2020 | 2021 |
|---|---|---|---|---|
| Mining | Explosives (t) | 111.52 | 112.89 | 151.70 |
|  | Diesel (t) | 213.19 | 215.80 | 290.00 |
| Crushing | Electricity (MWh) | 903.09 | 813.32 | 1131.20 |
| Calcination | Electricity (MWh) | 11,573.56 | 13,841.10 | 18,751.50 |
|  | Coal (t) | 46,445.90 | 46,490.70 | 66,427.99 |
| Output of quicklime (t) |  | 231,563.12 | 234,395.40 | 320,634.87 |

## 4. Results

### 4.1. Carbon Emissions Characteristics

Based on Formulas (2)–(9), the data in Table 2 were calculated and the carbon dioxide equivalent of each stage of producing one ton of quicklime at each production stage was obtained.

The amount of carbon emission generated from producing one ton of quicklime in Mine A ranges from 1096.68 kg $CO_2$ equiv. to 1176.96 kg $CO_2$ equiv., with 2.96 kg $CO_2$ equiv. in the mining stage, 3.07 kg $CO_2$ equiv. in the crushing stage and 1133.85 kg $CO_2$ equiv. in the calcination stage, on average. The results in Table 3 show that carbon emissions were stable in 2019 and 2020. The main reason is that from 2019 to the 2021, the coefficient of variation for the amount of material and energy used to produce lime, as well as the production of quicklime, was from 0.05% to 2.3%, or less than 15%. This means the production data of Mine A were stable in these three years.

Table 4 shows that the carbon dioxide emissions during the calcination stage are significant throughout the life cycle of lime production. The ultimate aim of using the IPCC emission factor method in the life cycle analysis of quicklime production is to accurately locate the key carbon emission sources and propose corresponding carbon reduction measures.

**Table 4.** Carbon emissions from each stage of production of one ton of quicklime from 2019 to 2021. (unit: kg $CO_2$ equiv.).

| Producing Stage | Mining | | Crushing | Calcination | | | Total |
|---|---|---|---|---|---|---|---|
| Carbon Sources | Explosives | Diesel | Electricity | Electricity | Coal | Decomposition of Limestone | |
| 2019 | 0.13 | 2.85 | 3.29 | 42.18 | 349.34 | 698.89 | 1096.68 |
| 2020 | 0.13 | 2.85 | 2.93 | 49.83 | 345.46 | 744.81 | 1146.01 |
| 2021 | 0.12 | 2.80 | 2.98 | 49.36 | 360.85 | 760.85 | 1176.96 |
| Average | 0.13 | 2.83 | 3.07 | 47.12 | 351.88 | 734.85 | 1139.88 |

Figure 2 shows the carbon emission structure distribution of quicklime from 2019 to 2021. During the life cycle of quicklime production, the decomposition of limestone at high temperatures is the main contributor of carbon emissions, which accounts for 64% of the total carbon emissions, on average. The second source is the traditional fossil fuel, coal, which accounts for 31% of the total carbon emissions, on average.

Based on the analysis of the carbon emission structure of quicklime production, we can propose possible low-carbon production measures through drawing upon strategies discussed by a range of scholars.

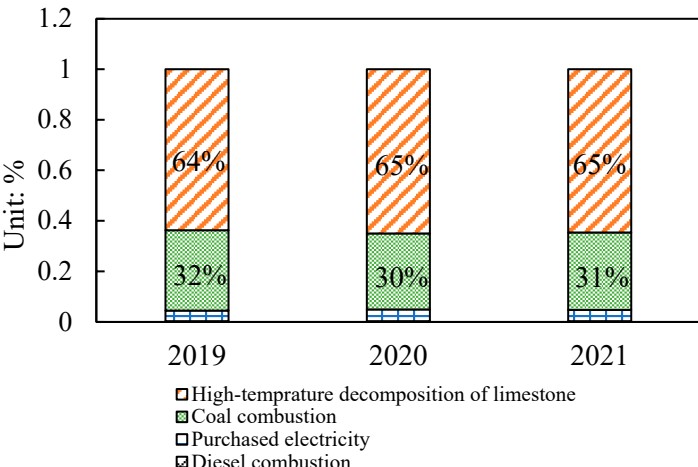

**Figure 2.** Carbon emission structure distribution of quicklime in the production process from 2019 to 2021.

(1) In the calcination stage, the high-temperature decomposition of limestone is the dominant factor of carbon emission. Post-combustion capture is a technology that can separate $CO_2$ from other components in the flue gas of the rotary kiln, thereby achieving $CO_2$ concentration. The technology is characterized by the installation of $CO_2$ capture devices at the tail of the exhaust gas, which does not require large-scale transformation of existing equipment. The absorption method is the most mature technology for $CO_2$ capture after combustion [32]. The methyldiethanolamine (MDEA) method is the most economical, and the $CO_2$ recovery rate (up to 99%) is the highest absorption method [32]. This method has been successfully applied to the Puguang gas field in China, and the mechanism of the method is that the natural gas is in contact with MDEA solution from bottom to top to decarbonize [33].

However, there is currently almost no application of the MDEA method within the mineral industry. One reason is that the lime industry faces particular technical challenges in applying this method. For example, high-temperature operation of lime kilns increases the complexity of carbon dioxide capture and may increase operational costs. Enterprises can gradually advance emission reduction measures according to their own operating conditions.

(2) In the calcination stage of quicklime production, coal combustion is another major source of carbon emission. Fuel optimization involves using clean energy to replace traditional fuels. Dudin et al. pointed out that natural gas is clean and energy efficient [34]. A Chinese cement company replaced all traditional fuel coal with natural gas by 2020, which not only saved energy consumption, but also reduced $CO_2$ emissions by 24% [35]. In this case study, if the traditional fuel coal was replaced with the natural gas in the calcination stage, the same production process can be maintained without a huge investment. According to the principle of equivalent substitution of fuel calorific value, referring to the general rule of comprehensive energy consumption calculation GBT2589-2020, the low calorific value of bituminous coal and natural gas is 19.57 GJ/t and 0.0389 GJ/m$^3$, respectively. Therefore, the calorific value of 1 t coal is equivalent to 503 m$^3$ natural gas. Based on the IPCC emission factor method, the carbon emission from 1 t coal combustion is 1.74 t, and the carbon emission from 503 m$^3$ natural gas combustion is 1.09 t. Replacing bituminous coal with natural gas, carbon emission can be reduced by 37%, indicating that natural gas can bring significant carbon reduction effects. According to lime production in 2022, the use of natural gas as a calcined fuel in the quicklime industry can reduce carbon emission by 120 million tons. In addition, straw is considered to be a promising green fuel because of its carbon neutrality. Through co-processing bituminous coal, 50% bituminous coal and 50% straw instead of 100% bituminous coal, the carbon emissions generated by fuel combustion can be reduced by 12% [36]. Mining operations may consider using clean

fuels or recycling straw, wood chips and other wastes to replace some traditional fuels in a synergistic manner, which will not only reduce carbon emissions from fuel combustion but also make full use of the calorific value of waste to achieve energy recycling.

### 4.2. Uncertainty Analysis Results

Uncertainty analysis is an important part of carbon emission accounting and is also the basis of judging the quality of accounting. In order to verify the difference between the carbon emission factors calculated in this study and those provided by the IPCC, the uncertainty of carbon emission factors for different energy sources and materials was calculated based on Formula (10); the $U_i$ of diesel, electricity, coal and quicklime is $\pm 1.57\%$, $\pm 17.86\%$, $\pm 6.11\%$ and $\pm 0.70\%$, respectively. Based on Formula (11), the results show that the overall uncertainty of the life cycle carbon emissions of quicklime for 2019, 2020 and 2021 are 2.13%, 2.07% and 2.09%, respectively.

Uncertainty about carbon dioxide emissions over the life cycle of quicklime production comes from many sources. The accuracy of the carbon emission accounting results in this study depends on the accuracy of the carbon emission factors. The carbon emission factors of lime are not uniform due to the different contents of calcium oxide and magnesium oxide in the limestone raw materials used by different enterprises; this is the reason for the difference in lime carbon emission factors. Similar uncertainties apply to coal's carbon emission factors. The fixed carbon content, volatile matter, ash and sulfur content of coal have a significant impact on the carbon emission factors of coal.

## 5. Discussion

### 5.1. Model Limitations

We found that the accuracy of the model was affected by differences in carbon emission factors. In the carbon emission structure of quicklime production, high-temperature decomposition of limestone is the major carbon emission source. In the carbon emission accounting of this process, the activity level and carbon emission factor of quicklime production are more important. The activity level is affected by the production efficiency of local enterprises. For example, the raw material limestone used in the production of quicklime ranges from 1.7 to 2 t. The carbon emission factor is affected by the content of calcium oxide and magnesium oxide in lime. The quality of lime produced in different areas is different, which affects the carbon emission coefficient of lime. Therefore, the IPCC model has geographical variance.

However, the availability of data also has an impact on the analysis of carbon emissions over the life cycle of lime production. Although the data of this study came from quicklime enterprises in the Yangtze River basin, the data of lime production are usually collected from software database and literature. In China, 87% of lime production enterprises are small enterprises (annual output < 100,000 t). These enterprises do not disclose the raw data of lime production. The limited availability of the data limits the data sources of other models and makes it impossible to compare the IPCC model with other models to further explore the accuracy of the IPCC model, which affects the analysis of carbon emissions in China's lime industry.

### 5.2. Synthesis of Results and Comparison with Other Studies

In our study, the carbon dioxide equivalent from producing one ton of quicklime was 1139.88 kg $CO_2$ equiv., on average. These carbon emissions are comparable to other studies, which have ranged between 0.98 t $CO_2$ equiv. to 1.26 t $CO_2$ equiv.5. The carbon emission factor of quicklime in this study was 0.71, while the carbon emission factor values of other studies ranged from 0.68 to 0.75. Different carbon emission factors result in different accounting results of carbon dioxide emissions.

We found that high-temperature decomposition of limestone was an important contributor to carbon emissions, accounting for an average of 64% of total carbon emissions. Fuel consumption was the second major source of carbon emission. These findings were

consistent with a previous study, which also concluded that two-thirds of the emissions were the decomposition of limestone at high temperature and 30% of carbon emissions came from fuel consumption6. In addition to making low-carbon improvements to important carbon sources, we will also make recommendations for energy conservation and carbon reduction in the next step for the production of the entire mining area, such as energy control.

## 6. Conclusions

1.  The carbon emission accounting boundaries for the life cycle of quicklime was determined, and its carbon emission sources were evaluated, including fuel (diesel, coal) combustion, explosion, electricity consumption and high-temperature limestone decomposition. The IPCC emission factor method was used to construct the life cycle carbon emission accounting model of quicklime and determined the carbon emission factor of producing unit quicklime of a lime enterprise in the Yangtze River basin.
2.  Regarding the Yangtze River Basin Enterprise's analysis, the amount of material and energy required to produce quicklime rose from 2019 to 2021. The largest energy consumption is coal, with an average coal consumption of 0.2 t per ton of quicklime produced. The carbon dioxide equivalent from producing one ton of quicklime ranged from 1096.68 kg $CO_2$ equiv. to 1176.96 kg $CO_2$ equiv. During the life cycle of lime production, the high-temperature decomposition of limestone was the largest carbon emission source, accounting for 64% of the total carbon emission. Coal combustion was the second carbon emission source, accounting for 31% of the total carbon emission.
3.  Low-carbon treatment suggestions were provided for the production of quicklime from two aspects. First, $CO_2$ capture technology is used to realize decarbonization of end flue gas. The second is to optimize the fuel. In the calcination stage, clean fuel or synergistic treatment of fuel can be used. In the open-pit mining stage, new energy equipment can be used instead of diesel equipment. It was suggested that the mining area can gradually promote carbon emission reduction measures according to its own operating conditions and move towards the road of green sustainable development.
4.  In order to verify the accuracy of the carbon emission in the life cycle of quicklime production, we calculated that the overall uncertainty of the life cycle carbon emissions of quicklime from 2019 to 2021 are 2.13%, 2.07% and 2.09%, respectively. The accuracy of the carbon emission accounting results in this study depends on the accuracy of the quicklime carbon emission factors and the coal carbon emission factors.

**Author Contributions:** E.W.: methodology, formal analysis, investigation, data curation and writing—original draft. Q.W.: conceptualization, methodology, writing—reviewing and editing and supervision. L.K.: resources, writing—reviewing and editing and supervision. G.Z.: resources and supervision. All authors have read and agreed to the published version of the manuscript.

**Funding:** This work was supported by the National Natural Science Foundation of China (41701624) and the Innovation and Entrepreneurship Project for college students in Hubei Province (S201910488071).

**Data Availability Statement:** The data of this study were acquired from the technical production department of Mine A.

**Acknowledgments:** We would like to express our gratitude to Simon Carter for his constructive comments, which helped improve our work a lot. The authors would also appreciate the anonymous reviewers for their contribution.

**Conflicts of Interest:** The authors declare no conflict of interest.

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
