# Peer review of "Study on Carbon Emission Characteristics and Emission Reduction Measures of Lime Production—A Case of Enterprise in the Yangtze River Basin"

_sustainability, doi:10.3390/su151310185_

Round 1
Reviewer 1 Report
This manuscript mainly studies the carbon emission characteristics and emission reduction measures of limestone production in a company located in the Yangtze River Basin, establishes a carbon life cycle emission calculation model for limestone production, and identifies key sources of carbon emissions. The manuscript puts forward the low-carbon transformation measures of the lime industry and suggestions for reducing carbon emissions, provides actual data on the carbon emission factor of the production unit of quicklime, and can be used as a reference to promote carbon neutrality in the industry. There are some issues to be addressed:
1. In 3.1 of this manuscript, in the description of the lime life cycle carbon emission accounting boundary map, one of the three stages is missing in the introduction of the carbon emission system boundary.
2. The location of the lime life cycle carbon emission accounting boundary map in 3.1 of this paper needs to be improved.
3. There were two citation errors in the process of citing the documents in 5.1 and 5.2 of this manuscript.
4. The position of Table 1 in this manuscript needs to be improved.
5. In part 2.1 of this manuscript, there are few cutting-edge research and development trend research on the carbon emission-related fields of the lime industry, and it needs to be added as appropriate.
6. The introduction of the IPCC model in Section 3.3 of this manuscript is too simple and does not declare the version of the IPCC model used.
7. During the application of the model, the author may consider verifying and revising the model to ensure the reliability and accuracy of the model.
8. In the analysis and discussion of the model's results, the author can further explore the limitations and deficiencies of the model to better understand the scope of application and constraints.
9. During the application and promotion of the model, the author may consider comparing and analyzing the model with other related models to better understand the strengths, weaknesses, and applicability of the model.
10. The manuscript can explore the significance and impact of the research results in more depth, to better explain the value and significance of the research, rather than simply listing the data
11. The references in the manuscript are insufficient and some are not accurate enough.
12. The data source introduction in part 3.4 of the manuscript introduces that the data for 2021 will be obtained in 2022. The author is requested to verify the time of data acquisition.
13. The data acquisition time is in July, but, the peak season of limestone production is usually in spring and summer, and the data acquisition frequency is less. And only one year's data is obtained, and long-term time-series data is not obtained. I suggest authors can obtain additional data for comparison. Combined with previous carbon emission reduction measures, more targeted emission reduction measures are put forward.
In conclusion, this manuscript's image, format, and content still need some adjustments, and I feel that this manuscript is more like a technical report than an academic research paper.
This manuscript mainly studies the carbon emission characteristics and emission reduction measures of limestone production in a company located in the Yangtze River Basin, establishes a carbon life cycle emission calculation model for limestone production, and identifies key sources of carbon emissions. The manuscript puts forward the low-carbon transformation measures of the lime industry and suggestions for reducing carbon emissions, provides actual data on the carbon emission factor of the production unit of quicklime, and can be used as a reference to promote carbon neutrality in the industry. There are some issues to be addressed:
1. In 3.1 of this manuscript, in the description of the lime life cycle carbon emission accounting boundary map, one of the three stages is missing in the introduction of the carbon emission system boundary.
2. The location of the lime life cycle carbon emission accounting boundary map in 3.1 of this paper needs to be improved.
3. There were two citation errors in the process of citing the documents in 5.1 and 5.2 of this manuscript.
4. The position of Table 1 in this manuscript needs to be improved.
5. In part 2.1 of this manuscript, there are few cutting-edge research and development trend research on the carbon emission-related fields of the lime industry, and it needs to be added as appropriate.
6. The introduction of the IPCC model in Section 3.3 of this manuscript is too simple and does not declare the version of the IPCC model used.
7. During the application of the model, the author may consider verifying and revising the model to ensure the reliability and accuracy of the model.
8. In the analysis and discussion of the model's results, the author can further explore the limitations and deficiencies of the model to better understand the scope of application and constraints.
9. During the application and promotion of the model, the author may consider comparing and analyzing the model with other related models to better understand the strengths, weaknesses, and applicability of the model.
10. The manuscript can explore the significance and impact of the research results in more depth, to better explain the value and significance of the research, rather than simply listing the data
11. The references in the manuscript are insufficient and some are not accurate enough.
12. The data source introduction in part 3.4 of the manuscript introduces that the data for 2021 will be obtained in 2022. The author is requested to verify the time of data acquisition.
13. The data acquisition time is in July, but, the peak season of limestone production is usually in spring and summer, and the data acquisition frequency is less. And only one year's data is obtained, and long-term time-series data is not obtained. I suggest authors can obtain additional data for comparison. Combined with previous carbon emission reduction measures, more targeted emission reduction measures are put forward.
In conclusion, this manuscript's image, format, and content still need some adjustments, and I feel that this manuscript is more like a technical report than an academic research paper.
Author Response
Dear reviewer,
Please check the attachment, thank you.

Reviewer 2 Report
Please check the attachment.

Minor editing of the English language is required.
Reviewer 3 Report
1. In the Introduction, the scientific contribution of the authors should be more emphasized.
2. Is Figure 1 made by the authors?
3. There is a "mess" in Table 2 when it comes to significant numbers.
4. Lack of page and line numbering makes reading and review difficult.
5. Could CO2 from burning fossil fuels be used in a cost-effective way to produce lime?
6. Is it planned to introduce CO2 storage method in lime technology?
Minor editing of English language required.
Author Response

(The authors gave the same response as above.)

Round 2
Reviewer 2 Report
Please check the attachment.

Minor editing in English is required.
Reviewer 3 Report
In this form the manuscript is ready for publication.